# Differential growth enhancement followed by notable microbiota modulation in growing-finishing pigs by *Bacillus subtilis* strains ps4060, ps4100, and a 50:50 strain mixture

**Jun Ho Song[1], Sung-Su Park[2], In Ho Kim[1]\*, Yangrae Cho[2]\***

1 Department of Animal Resource & Science, Dankook University, Cheonan, Republic of Korea,
2 Proxenrem, Osong Saengmyeong1-ro, Osong-eup, Chungju-si, Republic of Korea

\* yangchorae@gmail.com (YC); inhokim@dankook.ac.kr (IHK)

**Data Availability Statement:** The fecal microbiota sequencing data generated during the current study are available in the NCBI SAR repository

## Abstract

A 50:50 blend of two *Bacillus subtilis* strains positively impacted the productivity of finishing pigs. Given this observed effect, we hypothesized that each strain has distinct effects on weight gain and their influence on gut microbiota. In a 16-week test, 160 pigs were divided into four groups: basal diet, *B. subtilis* ps4100, *B. subtilis* ps4060, and 50:50 mixture supplemented. Subsequently, we compared body weight and fecal microbiota. Among the supplements, ps4100, ps4060, and the 50:50 mix yielded respective average daily weight gains (ADG) of 3.6%, 4.6%, and 3.9% by the 6th week. The weight difference was maintained through the 16th week. At the 11th week, the difference in α-diversity among the fecal microbiota was marginal, and 17 of 229 genera showed differential abundance between the control and either of the treatment groups. A total of 12 of the 17 genera, including *Lactobacillus*, showed differential abundance between the ps4100 and ps4060-fed groups, and only *Eubacterium* consistently decreased in abundance in both the ps4100 and ps4060 groups. In comparison, microbial diversity was significantly different at the 16th week ($p <$ 0.05), with 96 out of 229 genera exhibiting differential abundance. A total of 42 of the 96 genera exhibited similar patterns in both the ps4100 and ps4060 groups compared to the control group. Additionally, 236 of 687 microbial enzymes with differential abundance deduced from 16S rRNA reads showed similar differential abundance in both groups compared to the control group. We concluded that the overall microbial balance, rather than the dominance or significant decrease of a few specific genera, likely caused the enhanced ADG until the 11th week. Substantial changes in microbiota manifested at the 16th week did not cause dramatically increased ADG but were a consequence of weight gain and could positively affect animal physiology and health afterward.

(deposit number SUB14130364, BioProject accession number PRJNA1061231). All raw data such as raw body weight and feed amounts, will be available upon request to either of the corresponding authors, YC or IHK, or the Dankook Smart Animal Bioresearch Center (http://www.dankook.ac.kr/web/ins403, email: 12131722@dankook.ac.kr).

**Funding:** This work was partly supported by the Platform Technology Development Program (RS-2022-TI023566, S3309275), the Functional Food Program (RS-2023-00282826), and the Technology Development (R&D) Support Program (S3374815), and of the Ministry of SMEs and Startups (MSS, Korea) to YC. The funders had no role in study design, data collection and analysis, decision to publish, or preparation of the manuscript. No additional external funding was received for this study.

**Competing interests:** YC owns over 70% stocks of a startup company, ProxEnrem INC. This manuscript could facilitate the process of registering the strains described in the manuscript as animal diet supplement in the Republic of Korea. This does not alter our adherence to PLOS ONE policies on sharing data and materials.

## Introduction

Researchers in the field of animal science are actively seeking microbes that can enhance productivity in economically important animals. These probiotics offer various benefits when introduced into animal diets, including promoting health to alleviate challenges such as post-weaning diarrhea [1,2], boosting weight gain in finishing pigs [3], and serving as alternatives to antibiotics for promoting animal growth [1,4–7]. Additionally, they contribute to reducing environmental pollution while enhancing productivity [8].

Commonly used probiotics in animal feed include members of the genera *Lactobacillus*, *Saccharomyces*, and *Bacillus*, with spore-forming *Bacillus* demonstrating advantages in terms of stability during production, logistics, and efficacy [6,9,10]. Among the probiotics used, the spore forming bacterial strains are advantageous because spores can withstand low pH in the stomach and can endure bile salts and various digestive enzymes secreted in the gastrointestinal tract [10,11]. Spore forming strains, such as those of *B. subtilis*, *B. licheniformis*, *B. coagulans*, or their mixtures have shown beneficial effects associated with the modulation of the gut microbiome [12].

The gut microbiome is not only associated with enhancing animal productivity and weight gain but also with combating metabolic diseases linked to overweight and obesity [13–15]. Adequate probiotic consumption fosters gut health by promoting a balanced and diverse gut microbiome [16], aiming to prevent unhealthy weight gain. Specific probiotic feeding strategies have demonstrated success in improving productivity, potentially by increasing beneficial bacteria [8]. However, the relevance of changes in the microbiota to the weight gain of finishing pigs is unclear and has yet to be further investigated. Moreover, the cause-and-effect relationship between enhanced weight gain and the modulation of microbiota in finishing pigs has not been determined, and the health effects of weight gain in the context of unhealthy obesity are also not well understood. Ongoing research continues to explore the potential of probiotics in enhancing gut microbial diversity and their impact on overall health.

In a previous study, two *B. subtilis* strains were shown to enhance the productivity of finishing pigs without compromising carcass quality when fed a 50:50 mixture for 10 weeks [17]. Here, we individually fed these strains to investigate whether each strain acts differently in terms of productivity, meat quality, and interactions between the two strains in enhanced productivity. We hypothesized that (1) each *B. subtilis* strain (ps4100 and ps4060) would have distinct effects on the productivity and gut microbiota of finishing pigs, and (2) the enhanced productivity observed would be associated with specific modulations in the gut microbiota, including changes in microbial diversity and the relative abundance of pathogenic and beneficial bacteria. We tested whether enhanced productivity is associated with modulation of the fecal microbiota and the differential abundance of metabolic pathways predicted from the microbial community.

## Materials and methods

### IACUC approvals

The study was reported in accordance with ARRIVE guidelines (https://arriveguidelines.org" https://arriveguidelines.org) to the Animal Care and Use Committee (IACUC) at Dankook University, located in Cheonan, ROK. We used piglets produced internally in a sow room, which is capable of housing 30 piglets during farrowing, located at the university research pig farm in Jeonui, Chungnam. The committee has approved the protocols for animal experiments and management under approval number DK-1-2216. The Republic of Korea adheres to international standards and guidelines, which encompass the International Guiding Principles for

Biomedical Research Involving Animals and the OECD Principles of Good Laboratory Practice. These regulations govern animal care and use through oversight by the Ministry of Food and Drug Safety and the Ministry of Agriculture, Food and Rural Affairs.

### Two strains of the probiotic *B. subtilis*

The *Bacillus subtilis* strains ps4100 and ps4060 were obtained from the digestive tract of Korean rhinoceros beetle (*Protaetia brevitarsis* seulensis), which were sourced in the Republic of Korea. The genome sequences of these strains have been recorded in GenBank under the accession numbers CP081458 for ps4060 and CP076445 for ps4100. The screening procedure and their limited antibiotic resistance were detailed in a previous study [18]. The spore cells of the *Bacillus subtilis* ps4100 and ps4060 strains were separately produced following the methodology described in a prior study [17]. Subsequently, the spore suspension of each strain was absorbed into silica and air-dried at 80˚C, resulting in the final test materials (ELT Science, Osong, Korea). These end products were fed either individually or blended in a 50:50 mixture.

### Experimental design, animals, and housing

For our 16-week experiment, we utilized 160 growing pigs ([Landrace × Yorkshire] × Duroc) with an average body weight (BW) of 24.77 ± 2.73 kg and an age of approximately nine weeks. The pigs were allocated to four experimental groups based on their initial body weight using a stratified random block design. Each treatment consisted of 10 replication pens, each containing four pigs, two barrows and two gilts. Thus, a total of 20 barrows and 20 gilts were allotted to each treatment. The experimental groups were differentiated according to the following specific probiotic strains: (1) the control group (CON), which was fed the basal diet (S1 Table) with no supplements; (2) treatment group 1 (TRT1), which was fed the basal diet supplemented with ps4100; (3) treatment group 2 (TRT2), which was fed the basal diet supplemented with ps4060; and (4) treatment group 3 (TRT3), which was fed the basal diet supplemented with equal amounts of ps4100 and ps4060. The supplements were mixed with the basal diet to reach a probiotic concentration of $1 \times 10^9$ cfu per kg of the basal diet. The basal diet was formulated to meet the nutrient requirements for low-protein diets specified in the 2012 edition of the National Research Council (S1 Table). All pigs were housed in an environmentally controlled room with a plastic floor and had *ad libitum* access to food and clean water throughout the experimental period.

### Growth performance

To investigate the effect of probiotics on the growth of grower-finisher pigs, body weight was measured individually at the beginning of the experiment (week 0) and at the end of weeks 6, 11, and 16. The feed consumption per pen was determined by subtracting the remaining feed from the initial amount when the body weight was measured at the end of weeks 6, 11, and 16. The average daily gain (ADG) of body weight, average daily feed intake (ADFI), and gain-to-feed ratio (G:F) were estimated as the averages from four pigs co-housed and group-fed together. They calculated using the following formulas: [(final BW–initial BW)/total number of days], [(initial amount of diet–remaining amount of diet)/total number of days], and [ADG/ADFI]. The cumulative body weight change was calculated as $[[(\text{final } BW_E - \text{initial } BW_E) - (\text{final } BW_C - \text{initial } BW_C)]/(\text{final } BW_C - \text{initial } BW_C)]$. $BW_E$ and $BW_C$ respectively represent the body weight of the experimental and control groups.

## Digestibility, carcass grade, and meat quality

We measured digestibility, carcass grade, and meat quality as described in a previous publication [17]. Detailed protocols are described in S1 Appendix.

## Metagenomic sequencing of the fecal microbiome

Eighteen fecal samples were collected at the end of week 6, while 24 fecal samples were collected at the end of week 16. There were 6 individuals in the CON, TRT1, and TRT2 groups in both cases, and the TRT3 group was added at the latter week. A 100 mg portion was taken from each fecal sample to extract genomic DNA using the DNeasy PowerSoil Pro Kit from Qiagen (Qiagen, Palo Alto, CA, USA). Subsequently, the quantity and quality of the DNA were assessed using a Nanodrop from Thermo Fisher Scientific (Waltham, MA, USA) and a Bioanalyzer from Agilent (Santa Clara, CA, USA). DNA from the V3-4 region was amplified using a primer set (Illumina, San Diego, 16S Metagenomic Sequencing library Preparation kit, 16S V3-V4 forward primer, `TCGTCGGCAGCGTCAGATGTGTATAAGAGACAGCCTACGGG NGGCWGCAG`; 16S V3-V4 reverse primer, `GTCTCGTGGGCTCGGAGATGTGTATAAGAGACAG GACTACHVGGGTATCTAATCC`), and ~450 nucleotides were sequenced by DNA-Link (Seoul, ROK). The resulting product was purified and used as template DNA for subsequent amplification with a unique set of index primers for each template DNA following Illumina's protocol (San Diego, CA). This generated 42 indexed libraries, which were multiplexed and sequenced on an Illumina MiSeq platform, producing paired-end sequence reads. To perform quality trimming and sequence demultiplexing, we used in-house Perl scripts. Furthermore, noise, replicates, and chimeras were removed. Amplicon sequence variants (ASVs) were generated using Quantitative Insights Into Microbial Ecology (QIIME2, version 2023. 09) [19,20]. Q2-DADA2 was included in the QIIME2. The reads in the ASVs were assigned to operating taxonomic units (OTUs) using the classifier Silva (SSU138) and the 16S rRNA gene database [21,22], provided that they showed more than 99% sequencing identity.

## Fecal microbial community analyses

Alpha- and beta-diversity analyses were subsequently conducted on a rarefied ASV table utilizing the normalized sequence count of 45,703 ASVs. Richness and diversity indices, including observed ASVs, Chao1 estimates, Simpson's index, Shannon's index, evenness, and Faith's phylogenetic diversity, were calculated based on the rarefied ASV table. Principal coordinate analysis (PCoA) was performed on variance distance matrices to investigate the dissimilarity of the overall microbiota between the control group and three *Bacillus*-fed groups. The differential abundance of OTUs was analyzed by Gneiss [23] and ANCOM [24], which were incorporated in QIIME 2, along with ANOVA in the R package using normalized sequence counts [25] (Core R Team, 2021). The differential abundance of the gut microbiota was visualized using the pheatmap package in R (version 4.1.2). Functional predictions of enzymes and metabolic pathways, based on the Kyoto Encyclopedia of Genes and Genomes (KEGG) database, were performed using Phylogenetic Investigation of Communities by Reconstruction of Unobserved States (PiCRUST2) [26]. The KO results were subjected to KEGG pathway enrichment analysis using a custom Python script.

## Statistical analysis

We employed analysis of variance (ANOVA) incorporated in the R package (version 4.1.2) to analyze differences in OTUs, KOs, and metabolic pathways among the control group and the three treatment groups. A probability level of $p < 0.05$ was considered to indicate statistical

significance. We compared differences between all possible pairs using Tukey's multiple-comparison tests with Bonferroni correction. The Kruskal–Wallis test for alpha diversity and PERMANOVA for beta diversity were conducted in QIIME 2. PERMANOVA is a nonparametric multivariate statistical permutation test used to compare groups of objects, ensuring that the centroids and dispersion of the groups are equivalent for all groups. We utilized the Bray–Curtis distance and UniFrac distance methods.

## Results

### Impact on body weight increases and meat quality in pigs

In a 16-week experiment, supplementing grower-finisher pigs with *B. subtilis*—ps4100, ps4060, or a 50:50 mixture—significantly boosted weight gain compared to the basal diet ($p < 0.05$). Strain ps4060 led to the greatest increase (3.4%), followed by the mixture (2.8%) and ps4100 (2.1%) (Table 1). Any strain alone was sufficient for enhancing productivity in average daily gain (ADG) during the first six weeks up to the eleventh week ($p < 0.01$). While the 50:50 mixture initially trailed ps4060 but barely exceeded ps4100, it outperformed the average of both strains and had the highest ADG in the last five weeks (Table 1). The peak ADG occurred in the first 6 weeks, tapering off over time. The mixture's total weight gain matched the average weight gain of ps4100 and ps4060 by the end of the 16th week (Table 1). The supplementation also resulted in an enhancement of one of the meat quality indices, especially meat color (S1 Appendix).

### Modulation of fecal microbiota diversity

After being fed probiotics for 11 weeks, an average of 75,584 reads of bacterial 16S rRNA were generated from 18 fecal samples, representing approximately 150-day-old finisher stage.

**Table 1. The effect of dietary supplementation with *Bacillus subtilis* on the growth of growing-finishing pigs.**

| Items | CON | TRT1 | TRT2 | TRT3 | SEM | *p*—value |
|---|---|---|---|---|---|---|
| Body weight, kg | | | | | | |
| Initial | 24.8 | 24.8 | 24.8 | 24.8 | 0.22 | 1.000 |
| Week 6 | 53.7 | 54.8 (3.6%*) | 55.0 (4.5%*) | 54.8 (3.8%*) | 0.24 | 0.220 |
| Week 11 | 81.7[a] | 83.4[ab] (2.9%*) | 84.2[b] (4.3%*) | 83.5[ab] (3.2%*) | 0.31 | 0.037 |
| Week 16 | 113.7[a] | 115.6[ab] (2.1%*) | 116.7[b] (3.4%*) | 116.2[b] (2.8%*) | 0.36 | 0.015 |
| Initial—Week 6 | | | | | | |
| ADG, g | 689[a] | 714[b] (3.6%) | 721[b] (4.6%) | 716[b] (3.9%) | 4 | 0.011 |
| ADFI, g | 1654 | 1681 | 1685 | 1681 | 8 | 0.499 |
| G:F | 0.417 | 0.424 | 0.428 | 0.426 | 0.002 | 0.152 |
| Week 6—Week 11 | | | | | | |
| ADG, g | 800[a] | 818[ab] (2.3%) | 833[b] (4.1%) | 821[b] (2.6%) | 4 | 0.011 |
| ADFI, g | 2129 | 2132 | 2181 | 2152 | 12 | 0.429 |
| G:F | 0.376 | 0.384 | 0.382 | 0.382 | 0.002 | 0.756 |
| Week 11—Finish | | | | | | |
| ADG, g | 913 | 920 (0.8%) | 930 (1.9%) | 933 (2.2%) | 4 | 0.244 |
| ADFI, g | 2929 | 2900 | 2932 | 2903 | 22 | 0.935 |
| G:F | 0.361 | 0.368 | 0.368 | 0.369 | 0.001 | 0.1298 |

Abbreviations: CON, basal diet; TRT1, basal diet + ps4100; TRT2, basal diet + ps4060; TRT3, basal diet + 50:50 mixture of ps4100 and ps4060; SEM, standard error of means. The distinct superscripts denote significant differences ($p < 0.05$) among the means within the same row. An asterisk (*) indicates cumulative changes in body weight.

Similarly, after 16 weeks of probiotic feeding, an average of 84,954 reads were generated from 24 fecal samples, representing approximately 180-day-old finisher stage. Rarefaction curves indicated that the sequencing depth was sufficient to reasonably reflect the diversity of the microbiota (S1 Fig). The numbers of observed OTUs (± standard deviation) were 522.1 (± 48.4) and 504.0 (± 101.7) for the 11th and 16th-week samples, respectively. The OTU numbers in this study were similar to those reported in a 180-day longitudinal study spanning from the suckling to finisher stages [27].

*Small differences at the 11th week*: After 11 weeks of probiotic feeding, the microbiota showed no significant differences among the three groups in any of the indices except for the Shannon index (Fig 1A). Differences in the Shannon index were apparent between the ps4100 and ps4060 groups but not between the ps4100 and control groups. Principal coordinate analysis based on the Bray–Curtis distance matrix indicated differences among the three groups by the *11th* week (Fig 1B, $p = 0.03$). The group fed ps4100 was tightly clustered and separated from the control and ps4060 groups, which were dispersed (Fig 1B left). It appeared that ps4100 supplementation resulted in more alteration of the gut microbiota than ps4060.

*Large differences at the 16th week compared to the small differences at the 11th week*: For the control group, the OTUs remained similar between the 11th and 16th weeks (Fig 1A). The ps4100 group exhibited a significant increase in the number of observed features, which increased from 520.8 (±39.5) in the 11th week to 600.8 (±48.7) in the 16th week ($p < 0.01$). Conversely, the number of bacteria in the ps4060 group decreased from 509.7 (±35.8) to 415.2 (±122.1), although the intra-variation of bacterial numbers in each pig fed with ps4060 was high. There were more observed OTUs in the ps4100 group than in the ps4060 and the 50:50 mixture groups at the 16th week ($p < 0.01$; Fig 1A). Model-based estimates (ACE and Chao1), sum of phylogenetic branch length (Faith), and phylogenetic diversity (Simpson and Shannon) were also greater in the ps4100 group than in the ps4060 and 50:50 mixture groups ($p < 0.01$). Supplementation with ps4100 led to an increase in the number of OTUs, while supplementation with ps4060 or the 50:50 mixture resulted in a decrease in the number of bacteria that evenly spread across the bacterial domain. Bray–Curtis PERMANOVA also revealed a clear separation of the ps4100 group from the other groups (Fig 1B center and right).

## Comparison of taxa at the phylum level

The most dominant phylum, Firmicutes, comprised approximately 75% of the microbiota at both the 11th and 16th weeks, among a total of 30 detected phyla. The second most dominant phylum was Bacteroidota, accounting for 19% and 17% of the bacteria at the 11th and 16th weeks, respectively (Table 2 and S2 Fig). These two phyla collectively constituted more than 90% of the total gut microbiota in finishing pigs. The ratio of Firmicutes to Bacteroides tended to increase in the experimental groups at the 16th week compared to the 11th week, but the difference was not statistically significant (Table 2). Campylobacterota and WPS.2 were minor phyla but exhibited a decrease ($p < 0.05$) and an increase ($p < 0.01$), respectively, in the ps4060-fed group at the 11th week. At the 16th week, differential abundance of OTUs was evident across eight phyla, including Proteobacteria, Spirochaetota, and Euryarchaeota (Table 2).

## Comparison of taxa at the genus level

Initially, we used the Gneiss method to determine the general importance of the differential abundance of all genera among the experimental groups. One particular node indicated that the *Bacillus*-fed groups were different from the control group at the 11th week and that the control and ps4100 groups were different from the ps4060 or 50:50 mixture-fed groups at the 16th week (Fig 2A, green box). ANCOM identified one taxon at the 11th week, while five taxa

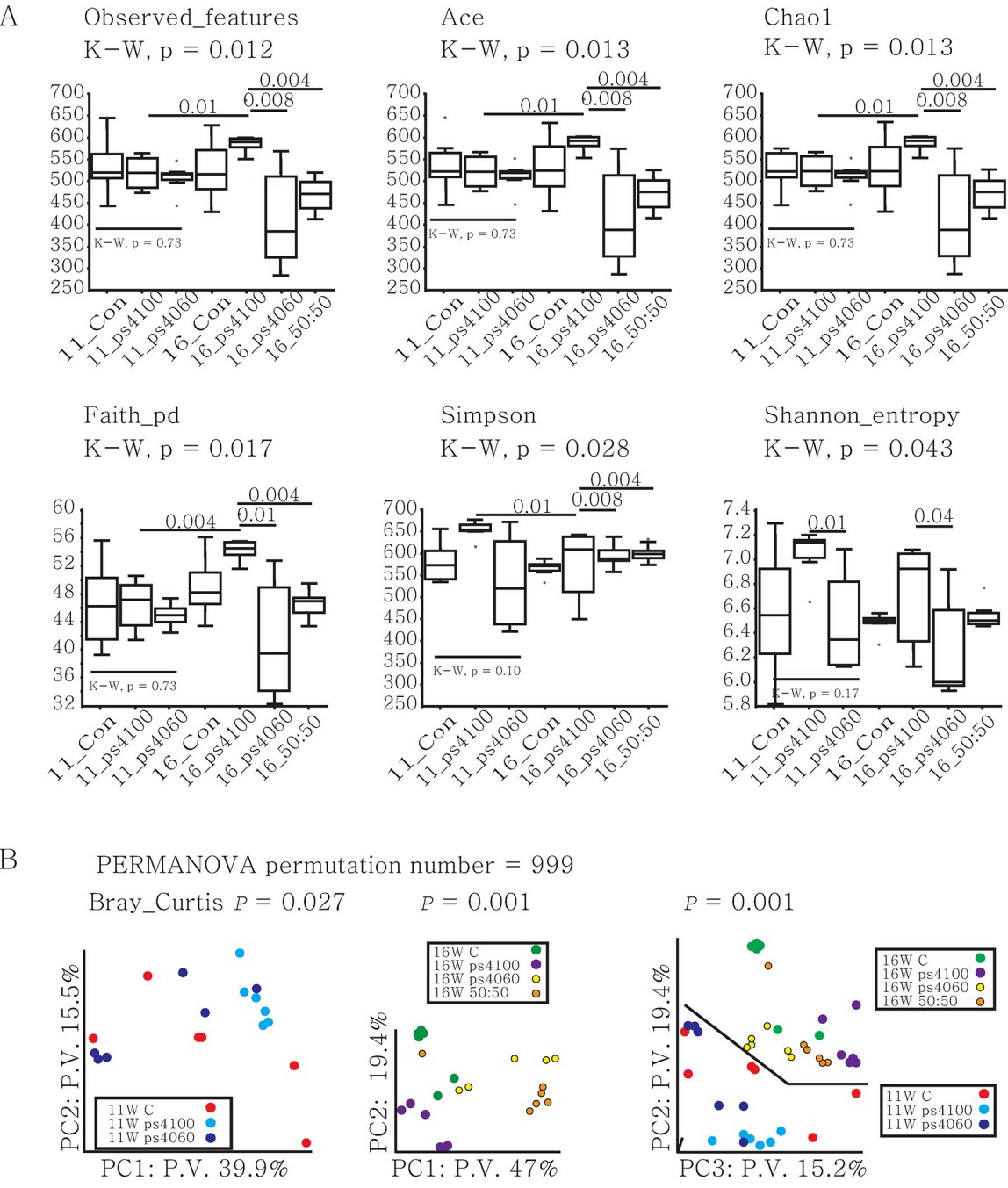

**Fig 1. Diversity of intestinal microbiota in the feces of finishing pigs fed a diet supplemented with or without *B. subtilis* probiotic strains**. Microbiota in the gut of 150-day-old finishing pigs after 11-week feeding trial and 180-day-old finishing pigs after 16-week feeding trial. A. Alpha-diversity indices. The numbers are *p*-values calculated by the Kruskal–Wallis test. B. Beta diversity was calculated by principal coordinate analysis. Abbreviations: P.V.: percent variation, K-W: Kruscal-Wallis.

at the 16th week (Fig 2B and 2C, red-marked taxa). Moreover, ANOVA revealed seventeen (6.9%) and ninety-six (41.9%) bacterial taxa with differential abundances among the 229 (133 ±13) genera at the 11th and 16th weeks, respectively (Fig 2B and 2C, Tables 3 and S2).

**Table 2. Comparison of OTU abundance among groups at the phylum level.**

| | 11th week | | | | 16th week | | | | |
|---|---|---|---|---|---|---|---|---|---|
| | Control | ps4100 | ps4060 | p-val | Control | ps4100 | ps4060 | 50:50 mix | *P*-val |
| Firmicutes | 35279±2242 | 33829±1932 | 29968±13007 | 0.48 | 33463±2319 | 34495±1605 | 34132±2941 | 33641±1284 | 0.83 |
| Bacterodota | 8716±1540 | 10130±1613 | 7667±3589 | 0.25 | 8294±1022 | 8058±1389 | 6815±1212 | 7078±638 | 0.08 |
| Ratio F/B | 4.0±1.0 | 3.3±0.7 | 3.9±0.7 | | 4.0±1.0 | 4.3±1.0 | 5.0±1.4 | 4.8±0.5 | 0.50 |
| Proteobacteria# | 285±114 | 237±150 | 119±97 | 0.09 | 1622±810[a] | 295±104[b] | 3618±2462 [a] | 3033 ±1444 [a] | 0.004 |
| Spirochaetota# | 438±669 | 327±481 | 117±84 | 0.51 | 1,436±495 [a] | 1,566±319 [a] | 345±91 [b] | 771±254[c] | 4.9.E-06 |
| Actinobacteriota | 283±124 | 420±110 | 388±313 | 0.49 | 153±126[a] | 348±258 [ab] | 229±40[ab] | 348±128 [b] | 2.0.E-03 |
| Planctomycetota# | 45±50 | 64±50 | 32±29 | 0.47 | 33±20 [a] | 206±73 [b] | 9 ±16[a] | 15±9 [a] | 3.7.E-08 |
| Verrucomicrobiota# | 18±27 | 9±7 | 1±2 | 0.22 | 26 ±15[a] | 84±13 [b] | 11 ±8[a] | 68±36[ab] | 1.1.E-05 |
| Fibrobacterota# | 19±16 | 13±7 | 15±9 | 0.64 | 22±21[ab] | 43 ±27[ab] | 9 ±6[a] | 42±23 [b] | 0.03 |
| Synergistota | 2±4 | 1±1 | 0±0 | 0.23 | 4±5[a] | 22±9 [b] | 0.4±1 [a] | 5±7 [a] | 1.4.E-05 |
| WPS-2* | 0.3±0.6 [a] | 0.0±0.0 [a] | 16±14.2 [b] | 0.006 | 1 ±1[a] | 12±17 [a] | 13 ±10[a] | 70±38 [b] | 6.2.E-05 |
| Campilobacterota* | 42± 28[a] | 32 ±24[a] | 3 ±4[b] | 0.02 | 199±114 | 145±83 | 72±48 | 177±109 | 0.12 |

Superscripts indicate significant differences in OTU abundance within each row ($p < 0.05$). An asterisk (*) and a hash (#) indicate phyla that showed differential abundance according to ANCOM at the 11th and 16th weeks, respectively.

At the 11th week, *Lactobacillus*, recognized as a health-promoting bacteria [28], was the most abundant genus in the control group (5,778±5,246, 12.7%). However, it was less in the ps4100 group (1,050±910, 2.3%) but more abundant in the ps4060 group (7,357±4,325, 16.2%) than the control group, resulting in a significant difference between the ps4100 and ps4060 groups (Fig 2B, $p < 0.05$). The other 11 of the 16 genera, including *WPS.2*, exhibited clearly different patterns of abundance between ps4100 and ps4060, while only *Eubacterium_siraeum* was decreased in both the ps4100 and ps4060 groups (Table 3). It was difficult to specify the causative microbes for the increased ADG.

By the 16th week, *Lactobacillus* remained the most abundant genus (5,876 out of 45,455, 12.9%) in the control group and in the *Bacillus*-fed groups, with no significant difference. The abundance of *Bifidobacterium*, which is renowned for its health-promoting properties by strengthening the mucosal barrier [29], increased in the ps4060 group and the 50:50 mixture group at the 16th week ($p < 0.0001$) but not at the 11th week. *Clostridium_sensu_stricto_1*, an opportunistic pathogen [30], was the second most abundant genus (4,620, 10.7%) in the control group at the 11th week and became the most abundant genus (5,877, 13.4%) by the 16th week. This genus exhibited an approximately 2-fold lower relative abundances in the ps4060-fed group compared to the control group. The *Bacteroides*, which is associated with sugar fermentation, obesity, and a possible marker of poor health [31], was also less abundant in the ps4060 and 50:50 mixture groups than the control group. Many genera, such as *Succinivibrio*, *Terrisporobacter*, *Treponema*, *Rikenellaceae*_RC9_gut_group, Bacteroidales_RF16_group, *Megasphaera*, *Selenomonas*, UCG-005, NK4A214_group, *Agathobacter*, and Prevotellaceae_NK3B31_group were abundant in the control group but not in the treatment groups. In contrast, less abundant genera, such as *Syntrophococcus*, *Tuzzerella*, *Erysipelotrichaceae*_UCG-009, and *Pseudoramibacter*, appeared to be more abundant in the experimental groups. A total of 96 out of 229 genera showed differential abundance among all groups. Of these, 42 genera exhibited similar patterns of abundance, while 54 genera showed different patterns of abundance between ps4100 and ps4060. The list of taxa displaying differential abundance among groups is available along with the corresponding read counts (S2 Table).

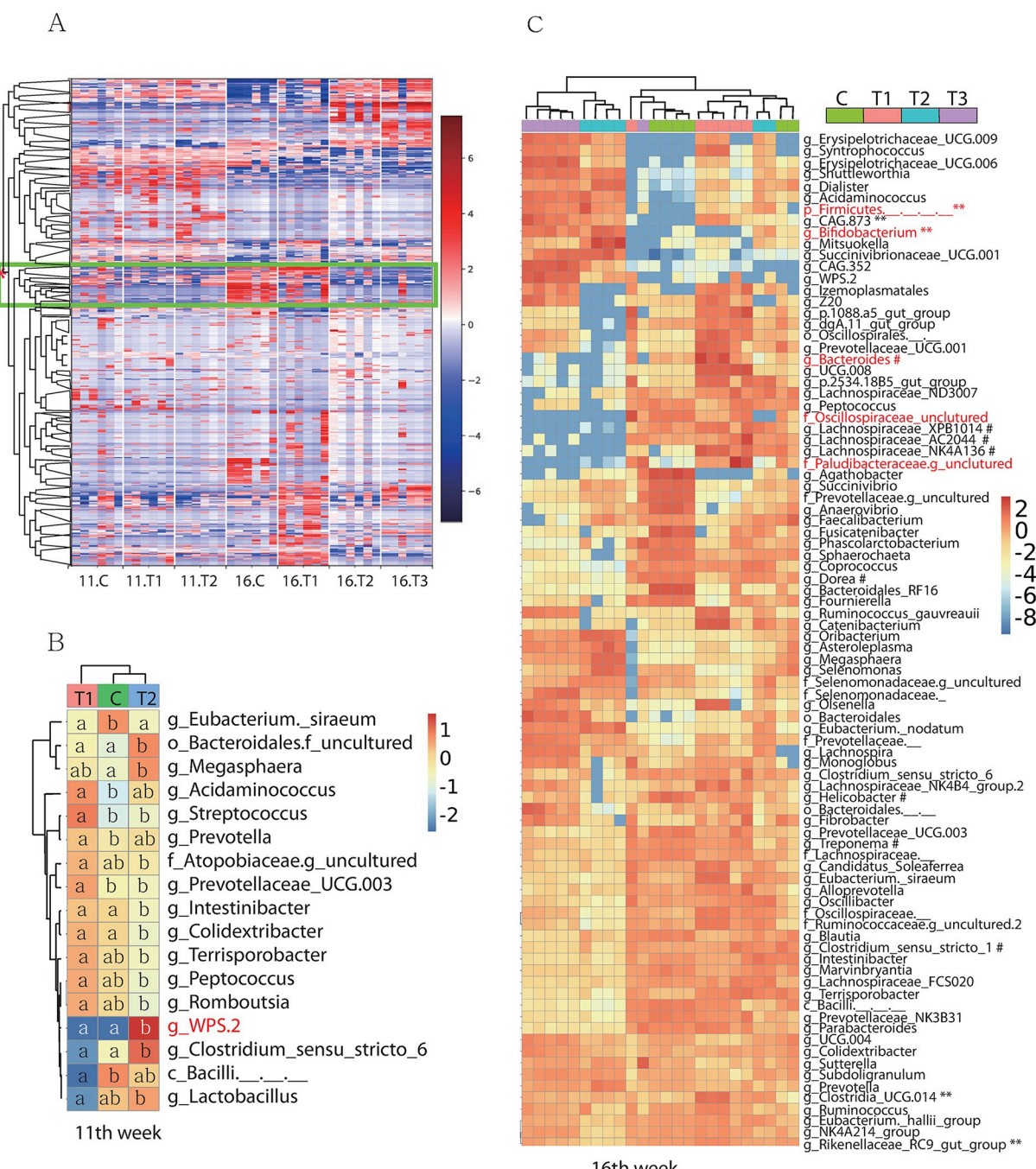

**Fig 2. Differential abundance of bacteria.** A. Graphical representation of the differential abundance of all bacterial genera analyzed by the Gneiss method. The green box with an asterisk (*) indicates a node showing a difference between the control and *Bacillus*-fed groups at the 11th week. B. Differential abundance of bacteria among groups at week 11, as determined by ANOVA. The letters in the colored squares indicate significant differences in OTU abundance within each row ($p < 0.05$). C. Differential abundance of bacteria among the experimental groups at week 16, as determined by ANOVA with Bonferroni correction. Those selected by ANCOM are marked in red. Two asterisks (**) and a hash (#) indicate the genera associated with health consequences.

The detection of the genera *Bacillus*, *Salmonella*, *E. coli*, and *Shigella* occurred at a low level, with an average of fewer than one read in all groups; thus, the differential abundance remained indistinguishable both at 11[th] and 16[th] weeks.

**Table 3. Genus-level differential abundance of bacteria among the experimental groups at the 11th week.**

| index | ANCOM | Control | TRT1 | TRT2 |
|---|---|---|---|---|
| g_Eubacterium_siraeum | | 164[b] | 71[a] | 68[a] |
| o_Bacteroidales | | 42[a] | 59[a] | 128[b] |
| g_Megasphaera | | 985[a] | 1325[ab] | 2629[b] |
| g_Acidaminococcus | | 43[b] | 153[a] | 96[ab] |
| g_Streptococcus | | 27[b] | 110[a] | 36[b] |
| g_Prevotella | | 3839[b] | 5997[a] | 4551[ab] |
| f_Atopobiaceae | | 33[ab] | 48[a] | 30[b] |
| g_Prevotellaceae_UCG.003 | | 103[b] | 206[a] | 91[b] |
| g_Intestinibacter | | 354[a] | 399[a] | 215[b] |
| g_Colidextribacter | | 106[a] | 132[a] | 61[b] |
| g_Terrisporobacter | | 2047[ab] | 2941[a] | 1375[b] |
| g_Peptococcus | | 112[ab] | 158[a] | 70[b] |
| g_Romboutsia | | 716[ab] | 1052[a] | 421[b] |
| g_WPS.2 | y | 0[a] | 0[a] | 17[b] |
| g_Clostridium_sensu_stricto_6 | | 92[a] | 30[a] | 280[b] |
| c_Bacilli_._._. | | 3229[b] | 310[a] | 2167[ab] |
| g_Lactobacillus | | 5778[ab] | 1050[a] | 7357[b] |

A total of 96 taxa were selected by ANOVA ($p < 0.008$). Distinct superscripts denote significant differences among the means within the same row as determined by Tukey's tests.

## Modulation of biochemical pathways

We performed Phylogenetic Investigation of Communities by Reconstruction of Unobserved States (PiCRUST) analysis to predict the functional relevance of the gut microbial community in relation to the enhanced weight gain based on the 16S rRNA gene sequences. The analysis identified a total of 22 Kyoto Encyclopedia of Genes and Genomes (KEGG) pathways and 413 KEGG orthology (KO) enzymes that differed among the three groups at the 11th week (Fig 3). Only ten of the 413 enzymes increased, while two decreased in both the ps4100 and ps4060 groups (Fig 3A). The biosynthetic pathways for tyrosine, phenylalanine, arginine, bacterial LPS, and thiamine diphosphate were slightly increased but the glucose and xylose degradation pathways decreased in both the ps4100 and ps4060 groups (Fig 3B). In comparison, out of 686 enzymes of differential abundance, 189 were increased and 47 decreased in both the ps4100 and ps4060 groups (Fig 3C). Among the 145 KEGG pathways showing differential abundance at the 16th week, five pathways showed over two-fold decreases in both the ps4100 and ps4060 groups compared to the control group. They included lipopolysaccharide (LPS) synthesis (PWY.7332), L-glutamate degradation (PWY.5088), and nucleotide degradation pathways (PWY.7198, PWY.7210, PWY.5705) (Table 4).

## Discussion

In the present study, *Bacillus subtilis* strains exhibited varying degrees of productivity enhancement in finishing pigs. The *B. subtilis* strain ps4060 demonstrated the highest efficacy, resulting in a 3.4% increase in productivity, followed by the 50:50 mixture (2.8%) and ps4100 (2.1%). The genus *Bacillus* was detected at low levels in all groups, possibly because the proportion of bacteria in the gut was relatively low. The feeding regimen of 1 x $10^9$ cfu/kg of feed constituted a relatively small proportion of the bacteria compared to the 1 x $10^{14}$ cfu bacteria within the gastrointestinal tract [32]. Although it is challenging to conclusively attribute the observed

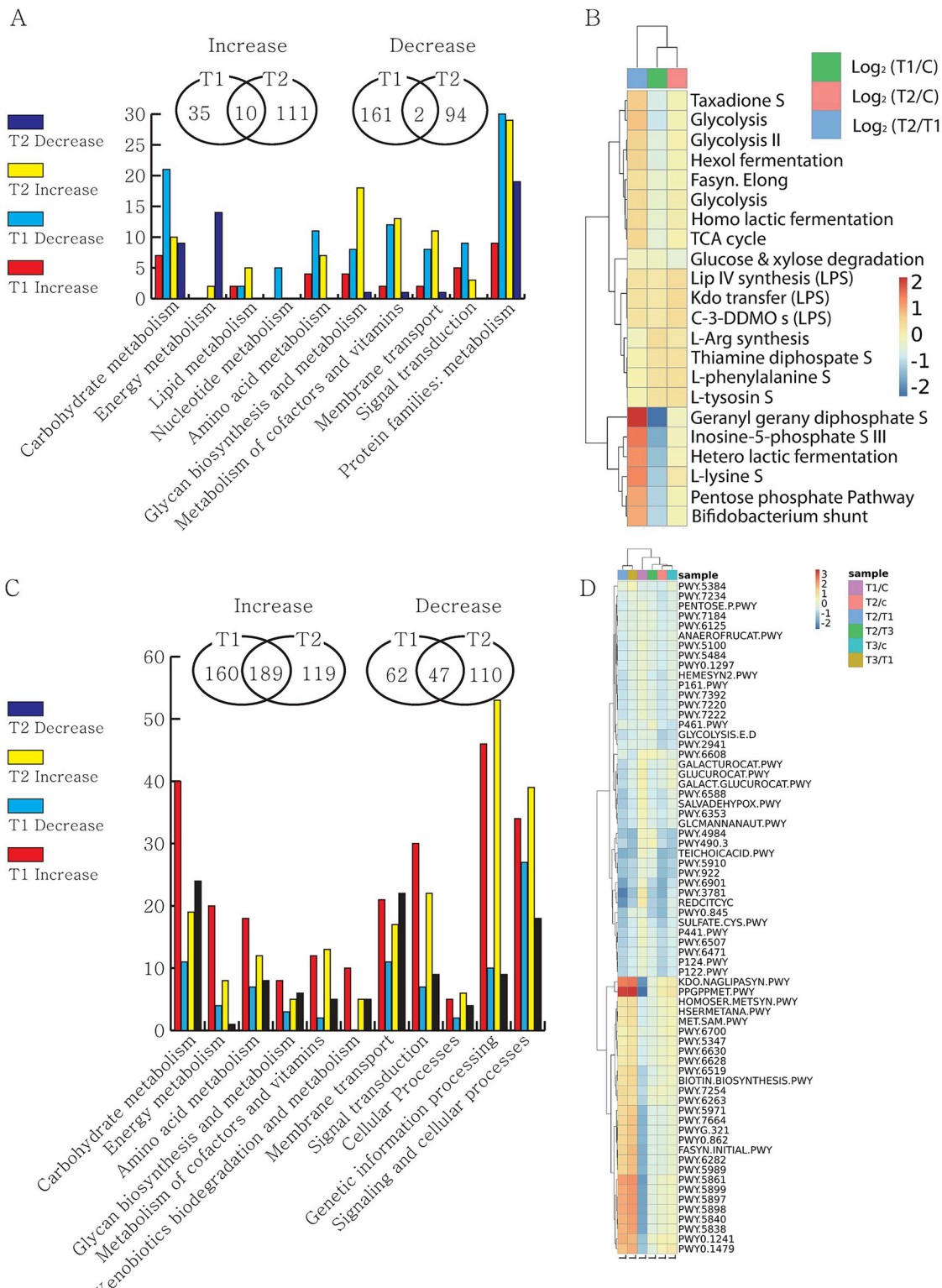

**Fig 3. PiCRUST prediction of the functional potential of a microbial community based on 16S rRNA gene sequencing data.** A. A total of 413 enzymes showing changes in expression at 11th week in the *Bacillus*-fed groups compared to the control group were categorized into KEGG pathways. B. Differential abundance of KEGG pathways shown by log2 ratios that were significantly changed in the *Bacillus*-fed groups at the 11th week. C. A total of 686 enzymes showing changes in expression at the 16th week in the *Bacillus*-fed groups compared to the control group were categorized into KEGG pathways. D. Differential abundance of KEGG pathways shown by log2 ratios that were significantly changed in the *Bacillus*-fed groups at the 16th week.

**Table 4. Differential abundance of selected KEGG pathways at the 16th week.**

| X.OTU.ID | 11th week | | | | 16th week | | | | |
|---|---|---|---|---|---|---|---|---|---|
| | Con | ps4100 | ps4060 | p-val | Con | ps4100 | ps4060 | 50:50 | p-val |
| PWY.5088 | 452 | 333 | 845 | 2.E-01 | 3,107 | 910 | 1,048 | 1,154 | 3.E-04 |
| PWY.7332 | 4,314 | 4,359 | 3,761 | 7.E-01 | 15,483 | 10,281 | 3,003 | 3,507 | 1.E-04 |
| PWY.7198 | 3,152 | 3,897 | 3,842 | 7.E-01 | 2,290 | 1,024 | 838 | 588 | 8.E-05 |
| PWY.7210 | 4,275 | 5,158 | 5,121 | 7.E-01 | 3,208 | 1,449 | 1,182 | 832 | 8.E-05 |
| PWY.5705 | 466 | 525 | 516 | 9.E-01 | 539 | 245 | 221 | 190 | 3.E-03 |

PWY.5088, L-glutamate degradation VIII (to propanoate); PWY.7332, superpathway of UDP-N-acetylglucosamine-derived O-antigen building blocks biosynthesis; PWY.7198, pyrimidine deoxyribonucleotides de novo biosynthesis IV, PWY.7210, pyrimidine deoxyribonucleotides biosynthesis from CTP; PWY.5705, allantoin degradation to glyoxylate III.

weight gain to the direct effect of the *Bacillus* strain, supplementation with strains ps4100 and ps4060 improved ADG. The 50:50 mixture of the two strains also improved the ADG and showed an average effect.

The study found a peak effect on weight and ADG at sixth week across all experimental groups, followed by a less increase at the 11th week the sixth week, with the gained weight maintained until the end of the 16th week. These results reinforce those of a previous 10-week feeding study, which demonstrated that *Bacillus* supplementation in a 50:50 mixture enhances pig productivity in principle [17]. This study further demonstrated the positive impact of this supplementation over approximately 10 weeks in growing pigs, as well as in finishing pigs, in a controlled laboratory setting. The effects were observed sooner and at a greater level in the growing pigs compared to the finishing pigs.

To understand the reasons underlying the enhanced ADG, we analyzed microbiota data to investigate whether probiotics reduce harmful bacteria [1,2] and increase beneficial bacteria [33]. No difference was apparent in the number of putative pathogenic bacteria, such as *Shigella*, *E. coli*, or *Salmonella*, between the control group and any of the *Bacillus*-fed groups in this study. This observation contrasts with other studies that demonstrated the positive impact of probiotics on harmful or beneficial bacteria. Those studies mostly focused on weaners or post-weaners [1,2,34], a crucial period during which the gut microbiota is established [27]. During this period of growth, the withdrawal of in-feed antimicrobial agents could increase mortality [35]. The negligible effect on the number of harmful bacteria in this study is probably because the intestinal microbiota in grower-finishers is more stable compared to the expanding microbiota during the weaner stage, which begins at the semisterile stage after birth [27]. Alternatively, the sequence reads may not have been deep enough to discern the differential abundance of low-frequency pathogens detected by PCR in other studies [1,2].

Next, we compared microbial diversity between experimental and control groups. If specific bacterial taxa were crucial for ADG improvement, they might have either increased or decreased in abundance in both the ps4100 and ps4060 groups compared to the control group. Thus, we searched for taxa exhibiting such trends. However, only *Eubacterium* showed consistent trends in the groups supplemented with either ps4100 or ps4060 at the 11th week. Instead, taxa that increased in the ps4100 group were decreased or unchanged in the ps4060 group, or vice versa, compared to the control group. For example, the abundance of the beneficial bacterium *Lactobacillus* decreased in the ps4100 group and increased in the ps4060 group, suggesting that its abundance is not necessary for the enhanced productivity. In summary, the difference in microbial diversity was small at the 11th week, when the *Bacillus*-fed groups had passed the peak of increased ADG compared to the control group ($p < 0.01$). The association

between any taxon and ADG increase was weak based on the differential abundance of individual microbial taxa. It implied that many microbial taxa with or without showing differential abundance contributed as a community to the enhanced ADG. Alternatively, low frequency microbiota played important roles but failed in detection due to the insufficient sequencing depth, as described in the previous paragraph.

In contrast to the 11[th] week, there was a significant difference in the microbiota at week 16. The fecal microbiota changed more in the later stages after ADG enhancement than in the early stages before it. This finding suggests that microbial diversity or species richness within the microbial community at week 16 was not the major cause of weight gain but rather a result of it. These changes might affect host physiology and play important roles in the gastrointestinal tract [36–38]. The low-level presence of the pathway PWY.7332 for the LPS synthesis in the ps4100, ps4060, and 50:50 mixture groups compared to the control group might be associated with the reduction of plasma LPS that our group previously reported [17]. The level of the LPS synthesis pathway, however, was unlikely to be the cause of ADG enhancement before 11 weeks of supplementation but might be important for protecting the liver and reducing systemic inflammation [39,40]. Thus, these changes contribute to animal health despite the weight gain. Similarly, the relative reduction of the L-glutamate degradation VIII pathway might be associated with decreased emission of ammonia and CO2 in the feces of the supplemented groups [17].

This study's results provide insight into a controversial issue of the efficacy of *Bacillus* supplementation for finishing pigs. Several studies on finishing pigs reported positive effects [8], while others indicated negative effects of the same probiotics [41]. The latter study reported the positive effects of early-stage treatment on body weight gain and the feed conversion ratio. As the feeding duration increased from the weaning stage to the finishing stage, the effects became more negative during the finisher stage. In the present study, the positive effect was significant during the 6 weeks of the grower stage and the following five weeks of the first half of the finishing stage but was marginal during the last five weeks of the finisher stage. In this study, probiotic supplementation experiment lasted for 16 weeks, which was started 6 weeks prior to the start of the finishing stage. Extended supplementation of probiotics showed similar trend of the decreased effect in other studies at the later stage [3,41]. In summary, *Bacillus* feeding enhanced body weight gain during the grower–finisher stage. Similarly, body weight enhancement was obvious during the finisher stage when supplementation was initiated at the end of the grower stage [17]. It is appropriate to interpret that the positive effects waned as the feeding period increased, as observed in the present study, rather than because the probiotics had negative effects on finishing pigs.

## Supporting information

**S1 Fig. Rarefaction curves.**
(DOCX)

**S2 Fig. Fecal microbiota structure based on 16S rRNA gene analysis.**
(DOCX)

**S1 Table. Composition of the experimental finishing pig diets (as-fed basis).**
(DOCX)

**S2 Table. Genus-level differential abundance of bacteria among the experimental groups at week 16.**
(DOCX)

**S1 Appendix. Digestibility, carcass grade and meat quality.**
(DOCX)

**S2 Appendix. Raw data for Tables 1–4.**
(DOCX)

## Acknowledgments

We thank Bongsek Kang and Sook-Jung Jeong for their assistance in maintaining and preparing *Bacillus* strains.

## Author Contributions

**Conceptualization:** In Ho Kim, Yangrae Cho.

**Data curation:** Jun Ho Song, Sung-Su Park, In Ho Kim, Yangrae Cho.

**Formal analysis:** Jun Ho Song, Sung-Su Park.

**Funding acquisition:** Yangrae Cho.

**Investigation:** Jun Ho Song.

**Methodology:** In Ho Kim.

**Project administration:** In Ho Kim, Yangrae Cho.

**Resources:** In Ho Kim.

**Software:** Sung-Su Park.

**Supervision:** In Ho Kim, Yangrae Cho.

**Validation:** In Ho Kim.

**Visualization:** Sung-Su Park.

**Writing – original draft:** Yangrae Cho.

**Writing – review & editing:** Yangrae Cho.

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
