## [Decision Letter · Decision Letter 0]

11 Jul 2024

PONE-D-24-23404Differential growth enhancement followed by notable microbiota modulation in growing-finishing pigs by *Bacillus subtilis* strains ps4060, ps4100, and a 50:50 strain mixturePLOS ONE

Dear Dr. Cho,

Thank you for submitting your manuscript to PLOS ONE. After careful consideration, we feel that it has merit but does not fully meet PLOS ONE’s publication criteria as it currently stands. Therefore, we invite you to submit a revised version of the manuscript that addresses the points raised during the review process.

We look forward to receiving your revised manuscript.

Kind regards,

Ewa Tomaszewska, DVM Ph.D

Academic Editor

PLOS ONE

“This work was partly supported by the Platform Technology Development Program (RS-2022-TI023566, S3309275) and Technology Development (R&D) Support Program (S3374815) of the Ministry of SMEs and Startups (MSS, Korea) to YC.”

“This work was partly supported by the Platform Technology Development Program (RS-2022-TI023566, S3309275) and Technology Development (R&D) Support Program (S3374815) of the Ministry of SMEs and Startups (MSS, Korea) to YC.”

“This work was partly supported by the Platform Technology Development Program (RS-2022-TI023566, S3309275) and Technology Development (R&D) Support Program (S3374815) of the Ministry of SMEs and Startups (MSS, Korea) to YC.”

“YC owns over 70% stocks of a startup company, ProxEnrem INC. This manuscript could facilitate the process of registering the strains described in the manuscript as animal diet supplement in the Repulic Korea.”

5. In this instance it seems there may be acceptable restrictions in place that prevent the public sharing of your minimal data. However, in line with our goal of ensuring long-term data availability to all interested researchers, PLOS’ Data Policy states that authors cannot be the sole named individuals responsible for ensuring data access (http://journals.plos.org/plosone/s/data-availability#loc-acceptable-data-sharing-methods).

6.Please review your reference list to ensure that it is complete and correct. If you have cited papers that have been retracted, please include the rationale for doing so in the manuscript text, or remove these references and replace them with relevant current references. Any changes to the reference list should be mentioned in the rebuttal letter that accompanies your revised manuscript. If you need to cite a retracted article, indicate the article’s retracted status in the References list and also include a citation and full reference for the retraction notice.

Additional Editor Comments:

Dear Authors,

there is a lack of a clear hypothesis.

with best regards

Ewa Tomaszewska

Reviewers' comments:

Reviewer's Responses to Questions

**Comments to the Author**

1. Is the manuscript technically sound, and do the data support the conclusions?

Reviewer #1: Yes

2. Has the statistical analysis been performed appropriately and rigorously? 

Reviewer #1: Yes

3. Have the authors made all data underlying the findings in their manuscript fully available?

Reviewer #1: Yes

4. Is the manuscript presented in an intelligible fashion and written in standard English?

Reviewer #1: Yes

5. Review Comments to the Author

Reviewer #1: The authors reported the differential growth enhancement followed by notable microbiota modulation in growing-finishing pigs by Bacillus subtilis strains ps4060, ps4100, and a 50:50 strain mixture.

The manuscript was relatively well prepared, and the methods and results were clearly presented.

It could be published in its current format, however, this reviewer suggests that the authors correct typos and errors throughout the manuscript.

6. PLOS authors have the option to publish the peer review history of their article (what does this mean?). If published, this will include your full peer review and any attached files.

Reviewer #1: No

---

## [Author Response · Author response to Decision Letter 0]

15 Jul 2024

1. PLOS One style: The style was checked and corrected accordingly following the instructions.

2. Funding Statement: This work was partly supported by the Platform Technology Development Program (RS-2022-TI023566, S3309275) and Technology Development (R&D) Support Program (S3374815) of the Ministry of SMEs and Startups (MSS, Korea) to YC. The funders had no role in study design, data collection and analysis, decision to publish, or preparation of the manuscript. There was no additional external funding received for this study. 

3. Acknowledgement: We removed the funding statement from the acknowledgement section. 

4. Competing Interests: YC owns over 70% stocks of a startup company, ProxEnrem INC. This manuscript could facilitate the process of registering the strains described in the manuscript as animal diet supplement in the Republic of Korea. This does not alter our adherence to PLOS ONE policies on sharing data and materials.

5. The Dankook Smart Animal Bioresearch Center (http://www.dankook.ac.kr/web/ins403, email: 12131722@dankook.ac.kr) is a non-author contact point for raw data. All data, such as body weight and feed amounts, will be available upon request to either of the corresponding authors, YC or IHK, or the non-author institute, Dankook Smart Animal Bioresearch Center, as indicated in the manuscript.

6. References were meticulously checked.

7. Lack of clear hypothesis: To explicitly describe testing hypothesis, we revised the last section of the last paragraph in the introduction section. “We hypothesized that (1) each B. subtilis strain (ps4100 and ps4060) would have distinct effects on the productivity and gut microbiota of finishing pigs, and (2) the enhanced productivity observed would be associated with specific modulations in the gut microbiota, including changes in microbial diversity and the relative abundance of pathogenic and beneficial bacteria. We tested whether enhanced productivity is associated with modulation of the fecal microbiota and the differential abundance of metabolic pathways predicted from the microbial community.”

---

## [Decision Letter · Decision Letter 1]

22 Jul 2024

Differential growth enhancement followed by notable microbiota modulation in growing-finishing pigs by *Bacillus subtilis* strains ps4060, ps4100, and a 50:50 strain mixture

PONE-D-24-23404R1

Dear Dr. Yangrae Cho,

We’re pleased to inform you that your manuscript has been judged scientifically suitable for publication and will be formally accepted for publication once it meets all outstanding technical requirements.

Kind regards,

Ewa Tomaszewska, DVM Ph.D

Academic Editor

PLOS ONE

Additional Editor Comments (optional):

Reviewers' comments:

Reviewer's Responses to Questions

**Comments to the Author**

1. If the authors have adequately addressed your comments raised in a previous round of review and you feel that this manuscript is now acceptable for publication, you may indicate that here to bypass the “Comments to the Author” section, enter your conflict of interest statement in the “Confidential to Editor” section, and submit your "Accept" recommendation.

Reviewer #1: All comments have been addressed

2. Is the manuscript technically sound, and do the data support the conclusions?

Reviewer #1: Yes

3. Has the statistical analysis been performed appropriately and rigorously? 

Reviewer #1: Yes

4. Have the authors made all data underlying the findings in their manuscript fully available?

Reviewer #1: Yes

5. Is the manuscript presented in an intelligible fashion and written in standard English?

Reviewer #1: Yes

6. Review Comments to the Author

Reviewer #1: The authors have appropriately addressed my previous concerns.

7. PLOS authors have the option to publish the peer review history of their article (what does this mean?). If published, this will include your full peer review and any attached files.

Reviewer #1: No

---

## [Editor Report · Acceptance letter]

30 Aug 2024

PONE-D-24-23404R1 

PLOS ONE

Dear Dr. Cho, 

I'm pleased to inform you that your manuscript has been deemed suitable for publication in PLOS ONE. Congratulations! Your manuscript is now being handed over to our production team.

Kind regards, 

on behalf of

Professor Ewa Tomaszewska 

Academic Editor

PLOS ONE